# Greater than pH 8: The pH dependence of EDTA as a preservative of high molecular weight DNA in biological samples

**Mia L. DeSanctis**[☯], **Elizabeth A. Soranno**[iD], **Ella Messner, Ziyu Wang, Elena M. Turner, Rosalia Falco**[iD][☯], **Hannah J. Appiah-Madson**[iD][☯], **Daniel L. Distel**[iD]*

Ocean Genome Legacy Center, Marine and Environmental Sciences, Northeastern University, Nahant, Massachusetts, United States of America

☯ These authors contributed equally to this work.

* d.distel@northeastern.edu

**Data Availability Statement:** All specimen data are available from the Ocean Genome Legacy Center specimen database (http://ogl.northeastern.edu/catalog; accession numbers S32101–S32150).

## Abstract

Ethylenediaminetetraacetic acid (EDTA) is a divalent cation chelator and chemical preservative that has been shown to be the active ingredient of the popular DNA preservative DESS. EDTA may act to reduce DNA degradation during tissue storage by sequestering divalent cations that are required by nucleases naturally occurring in animal tissues. Although EDTA is typically used between pH 7.5 and 8 in preservative preparations, the capacity of EDTA to chelate divalent cations is known to increase with increasing pH. Therefore, increasing the pH of EDTA-containing preservative solutions may improve their effectiveness as DNA preservatives. To test this hypothesis, we stored tissues from five aquatic species in 0.25 M EDTA adjusted to pH 8, 9, and 10 for 12 months at room temperature before DNA isolation. For comparison, tissues from the same specimens were also stored in 95% ethanol. DNA extractions performed on tissues preserved in EDTA pH 9 or 10 resulted in as great or greater percent recovery of high molecular weight DNA than did extractions from tissues stored at pH 8. In all cases examined, percent recovery of high molecular weight DNA from tissues preserved in EDTA pH 10 was significantly better than that observed from tissues preserved in 95% ethanol. Our results support the conclusion that EDTA contributes to DNA preservation in tissues by chelating divalent cations and suggest that preservative performance can be improved by increasing the pH of EDTA-containing DNA preservative solutions.

## Introduction

Ethylenediaminetetraacetic acid (EDTA) is an effective chelator of divalent cations that is widely used as a preservative in food [1, 2] and cosmetics [3, 4]. However, its utility as a preservative for high molecular weight (HMW) DNA—here defined as DNA fragments greater than 10 kb—in tissue samples has only recently been recognized [5]. Interestingly, EDTA has long been used for DNA preservation in tissues without explicit recognition. EDTA is one of three

DNA sequences were deposited in the Barcode of Life Data System database (BOLD; www.boldsystems.org) under accession numbers EDTA001-21 through EDTA044-21. All relevant data are within the paper and its Supporting Information files.

**Funding:** This work was funded by the Don Comb Research Scholarship (DLD), Ocean Genome Legacy Operations Fund (DLD), the Francis Goelet Charitable Lead Trust (DLD), and the National Science Foundation, Division of Biological Infrastructure (DBI 1722553 to Northeastern University). The funders had no role in study design, data collection and analysis, decision to publish, or preparation of the manuscript.

**Competing interests:** The authors have declared that no competing interests exist.

components of the preservative commonly referred to as DMSO-salt or DESS, which contains 20% dimethyl sulfoxide (DMSO) and 0.25 M EDTA dissolved in a saturated solution of sodium chloride and adjusted to a pH between 7.5 and 8 [6]. Tissue samples stored in this preservative often yield DNA that is comparable in quality and quantity to those preserved in ethanol or by cryopreservation [6–8]. DESS is effective at preserving DNA in tissues for a wide variety of organisms including microbial communities [8, 9], invertebrates [5, 10–12], fish [13, 14], birds [6], and mammals [7, 15–18]. DESS preserves DNA in tissues at room temperature, is simple and inexpensive to prepare, has low toxicity and flammability, and is not considered hazardous for shipping and transportation; this combination of features has contributed to its widespread use [5, 6, 10, 19].

While the effectiveness of DESS as a preservative has been most frequently attributed to its eponymous ingredient, DMSO, a recent investigation suggested that EDTA is the sole active ingredient with respect to preservation of HMW DNA [5]. In that study, tissues of three invertebrate taxa (*Mytilus edulis*, blue mussel; *Faxonius virilis*, virile crayfish; *Alitta virens*, clam worm) were stored for up to six months in each of the following solutions: (a) DESS, (b) one of the three ingredients of DESS individually, (c) all possible combinations of two of the three ingredients, or (d) 95% ethanol (EtOH). Among DESS and treatments containing DESS components, only those containing EDTA were effective at preventing degradation of HMW DNA. Additionally, for most taxa and time intervals, EDTA-containing treatments performed as well or better than 95% EtOH. Notably, all treatments, including 95% EtOH, performed poorly for *A. virens*. These results suggest that EDTA is an effective preservative of HMW DNA in some, but not all, tissues and taxa, and may be an alternative to DESS or 95% EtOH in some applications.

The effectiveness of EDTA as a preservative of HMW DNA in tissues has been attributed to its ability to effectively chelate $Mg^{2+}$ and $Ca^{2+}$ ions [5, 6], which are required as cofactors for most nucleases commonly found in biological tissues [20]. For example, DNase I, a ubiquitous endonuclease, requires $Mg^{2+}$ and $Ca^{2+}$ ions as co-factors to cleave the phosphodiester backbone of DNA [21]. Additionally, EDTA may be able to chelate and remove trace metal ions such as $Fe^{3+}$ and $Cu^{2+}$ that can greatly increase oxidative damage to DNA due to the production of hydroxyl radicals via Fenton-type reactions [22].

Although EDTA chelates a broad range of metal ions, it is only effective when all four of its carboxyl groups are deprotonated [23]. The $pK_a$ of the fourth carboxyl group of EDTA is 10.34 [23]. Therefore, while EDTA solutions can chelate metal ions at pH 8, their capacity to do so increases exponentially over the range of pH 8 to pH 10. Consequently, the effectiveness of EDTA in preventing the degradation of HMW DNA in biological tissues might also be expected to increase with increasing pH. Here we test this hypothesis by storing tissues of five aquatic taxa in 0.25 M EDTA solutions adjusted to pH 8, 9, or 10 for 12 months and evaluate the percent recovery (%R) and normalized yield (nY) of HMW DNA for each treatment. We compare these values to DNA extracted from fresh tissue and tissues preserved in 95% EtOH. Our aim in this investigation is to determine whether the performance of EDTA as a preservative of DNA in tissue can be improved by increasing pH. If so, this simple modification could be of value for research and commercial applications.

## Materials and methods

### Ethics statement

This study was carried out in accordance with the recommendations in the Guide for the Care and Use of Laboratory Animals of the National Institutes of Health and Northeastern University's Institutional Animal Care and Use Committee policies. Although no live vertebrate

animals were used in this study, the general principles of humane animal care were applied to the live invertebrate animals used.

## Tissue storage solutions

Storage solutions consisted of 0.25 M EDTA buffered with 10 mM Tris and adjusted to pH 8, 9, or 10 by addition of sodium hydroxide. Specifically, solutions were prepared by combining 125 mL of 0.5 M EDTA adjusted to pH 8, 9, or 10 with 50 mL of 50 mM Tris adjusted to pH 8, 9, or 10 and bringing the final volume to 250 mL with MilliQ water. All solutions were checked with a pH probe to ensure that they were the desired pH.

## Taxon selection and sourcing

Tissues from five taxa were examined in this study: *Alitta virens* (clam worm; $N = 10$), *Faxonius virilis* (virile crayfish; $N = 10$), *Homarus americanus* (American lobster; $N = 10$), *Scomberomorus maculatus* (Spanish mackerel; $N = 10$), and *Mercenaria mercenaria* (quahog; $N = 10$). *Alitta virens*, *F. virilis*, *H. americanus*, and *S. maculatus* specimens were purchased from Al's Bait & Tackle (Beverly, MA), A.J.'s Bait & Tackle (Meredith, NH), Ipswich Shellfish Fish Market (Ipswich, MA), and Pine River Fish Market (Revere, MA), respectively. *Mercenaria mercenaria* were donated by Cape Cod Shellfish & Seafood Co. (Boston, MA). Samples of all specimens were deposited into the Ocean Genome Legacy Center [24] specimen collection and can be accessed using specimen IDs S32101–S32150 [25].

## Dissection and tissue sampling

Live specimens of *A. virens*, *F. virilis*, *H. americanus*, and *M. mercenaria* were stored on ice prior to dissection. Immediately following euthanasia via spiking, five body, tail, abdominal muscle, and mantle tissue samples were collected from ten specimens of *A. virens*, *F. virilis*, *H. americanus*, and *M. mercenaria*, respectively. Ten *S. maculatus* specimens were purchased postmortem and were kept on ice until five muscle tissue samples were obtained from each specimen. Four tissue samples from each specimen of all taxa were placed in 1.8 mL cryotubes containing 1 mL of EDTA adjusted to pH 8, 9, or 10, or 95% EtOH, such that each treatment had 10 biological replicates per taxa. The cryotubes were stored at room temperature for 12 months (Fig 1). DNA was immediately extracted from the fifth tissue sample of the same individuals without preservation, hereafter referred to as fresh tissue. The DNA from these fresh tissue samples was visualized by gel electrophoresis immediately after extraction and then stored at -80˚C for 12 months, at which time DNA from all treatments, including fresh tissue, were analyzed concomitantly.

## DNA extraction and quantification

After 12 months of room temperature storage, each tissue sample was removed from its storage solution. Approximately 25 mg (avg. 26.70 ± 6.23 mg) of each tissue was subsampled for DNA extraction using the Qiagen DNeasy Blood and Tissue Kit (Hilden, Germany) according to the manufacturer's recommended protocol with the following modifications: all samples were treated with 4 µL of RNase A (100 mg/mL; Qiagen; Hilden, Germany) after digestion; DNA was eluted by adding two sequential 50 µL volumes of Buffer AE for a total elution volume of 100 µL.

Extracted DNA was visualized using agarose gel electrophoresis. Three µL of each DNA extract from *A. virens*, *F. virilis*, and *M. mercenaria* (avg. 0.1071 ± 0.0995 µg DNA) and 6 µL of each DNA extract from *H. americanus* and *S. maculatus* (avg. 0.0213 ± 0.0219 µg DNA) were

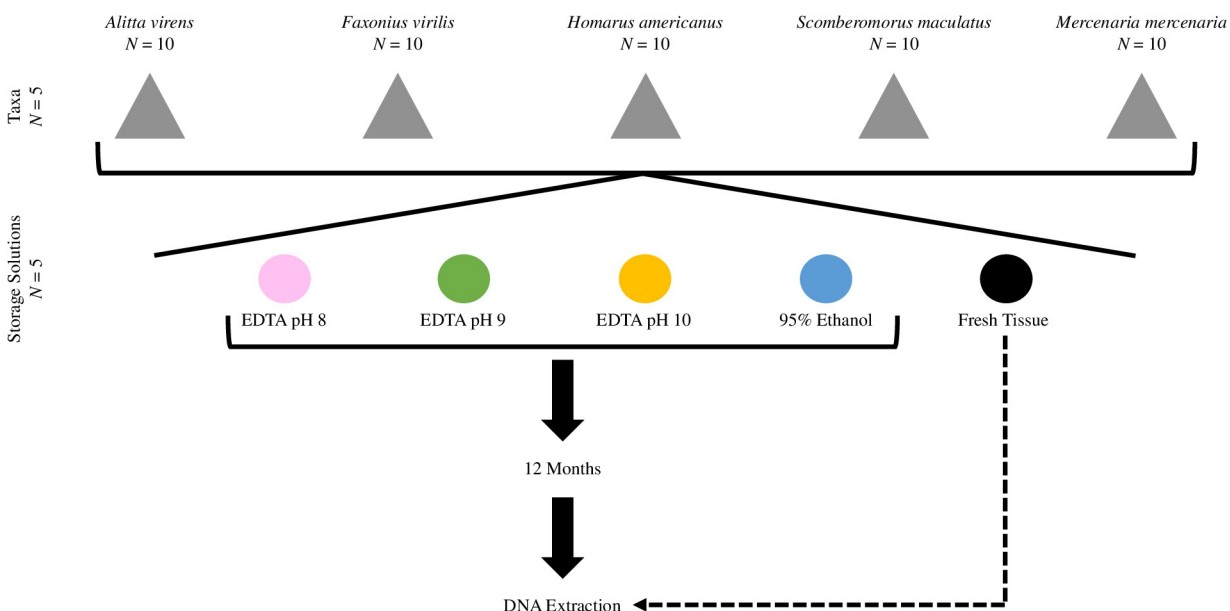

**Fig 1. Experimental design.** Tissues from five taxa were each exposed to four storage solutions for 12 months, after which DNA was extracted, analyzed, and compared to DNA extracted from fresh tissue.

loaded into a 16 cm horizontal slab gel (0.8% agarose, 1x TAE buffer containing 1% GelRed nucleic acid gel stain (Biotium; Fremont, CA)). The first and last lanes of each gel were loaded with either 0.33 μL or 0.66 μL of Quick Load Purple 1 kb Plus DNA Ladder (100 μg/mL; New England Biolabs; Ipswich, MA). Gels were run at 5 volts/cm for 50 minutes and visualized using a BioRad Gel Doc XR+ Molecular Imager (Hercules, CA).

Quantitative analyses of DNA extracts were performed using the Agilent Technologies TapeStation 2200 DNA Analyzer using the Agilent DNA Analyzer Genomic DNA Screen-Tapes and TapeStation Analysis Software version A.02.02 (SR1; Santa Clara, CA). One μL of each DNA extract (avg. 0.0243 ± 0.0296 μg DNA) was analyzed to determine both the percent recovery of HMW DNA (%R) and normalized HMW DNA yield (nY).

## PCR and DNA barcoding

The mitochondrial cytochrome oxidase I (*COI*) barcoding region was amplified by PCR from DNA extracts using the primers LCO1490_t1 (5′-TGTAAAACGACGGCCAGTGGTCAA CAAATCATAAAGATATTGG-3′) and HCO2198_t2 (5′-CAGGAAACAGCTATGAC TAAACTTCAGGGTGACCAAAAAATCA-3′) [26] for *A. virens*, *F. virilis*, *H. americanus*, and *M. mercenaria* and FISHCOILBC_ts (5′-CACGACGTTGTAAAACGACTCAACYAAT CAYAAAGATATYGGCAC-3′) and FISHCOIHBC_ts (5′-GGATAACAATTTCACACAG GACTTCYGGGTGRCCRAARAATCA-3′) [27] for *S. maculatus*. Each PCR amplification contained 2 μL DNA template, 17.5 μL OneTaq 2X Master Mix (New England Biolabs; Ipswich, MA), 10 μM of both forward and reverse primers, and was brought to 35 μL total volume with MilliQ water. PCR thermocycler conditions were initiated with a heated lid at 94˚C for 30 seconds, followed by 34 or 30 cycles of 94˚C for 30 seconds, 45˚C for 50 seconds or 52˚C for 40 seconds, and 68˚C for 60 seconds, with a final extension at 68˚C for 5 minutes for invertebrate taxa or *S. maculatus*, respectively, using a PCT-200 thermocycler (MJ Research, Inc.; Waltham, MA). PCR success was visualized by 0.8% agarose gel electrophoresis (described above).

The *COI* barcoding region was sequenced for PCR amplicons from fresh tissue and tissues stored in EDTA pH 10 and 95% EtOH for 12 months from three randomly selected specimens of each taxon. As in Sharpe, Barrios [5], 10 μL of each selected amplicon was bi-directionally Sanger sequenced on an Applied Biosystems 3730xl DNA Analyzer (Foster City, CA) at GEN-EWIZ (South Plainfield, NJ). Resulting sequences were edited and analyzed using Geneious v.8 (Auckland, New Zealand), automatically trimming ends to remove sequence with greater than a 1% chance of error per base and setting 500 bp as a minimum threshold for a successful read. Assembled contigs were deposited in the Barcode of Life Data System (BOLD [28]) database under accession numbers EDTA001-21 through EDTA044-21.

## Statistical analysis

In our analyses, DNA fragments greater than 10 kb were considered HMW DNA. To estimate the quantity of HMW DNA, we first used the Agilent Technologies TapeStation 2200 DNA Analyzer (Santa Clara, CA) to evaluate the quantity of low molecular weight DNA (150 bp to 10 kb) and the total amount of DNA present in each sample. Then we calculated the quantity of HMW DNA for each sample by subtracting the low molecular weight DNA from the total amount of DNA present. Finally, we determined the percent of HMW DNA recovered (%R) using the following formula:

$$\%R = \frac{\frac{ng}{\mu l}\ HMW\ DNA(> 10\ kb)}{\frac{ng}{\mu l}\ total\ DNA} * 100\%$$

Normalized HMW DNA yield (nY) was calculated as follows:

$$nY = \frac{\mu g\ HMW\ DNA(> 10\ kb)}{mg\ extract\ tissue\ weight}$$

Data for each taxon were analyzed separately in RStudio version 1.4.1717 [29, 30]. A Shapiro-Wilk normality test was used to determine if the data were normally distributed. If data were not normally distributed, a non-parametric Friedman $\chi^2$ followed by a Bonferroni-corrected Friedman Conover post-hoc test was used to test the effect of storage solution on DNA quality using the "PMCMR" package version 4.3 [31]. If the data were normal, a one-way repeated measures ANOVA followed by a Bonferroni-corrected Tukey post-hoc test was performed with the "nlme" package [32].

## Results

DNA was recovered from all taxa, samples, and treatments after 12 months of storage, with the exception of two *F. virilis* samples that could not be recovered from EDTA pH 8 for technical reasons. These two specimens were excluded from all quantitative analyses. %R across all taxa and treatments ranged from 5.5–76.1% and nY ranged from 0.0007–0.2431 μg DNA/mg tissue. Statistical models were significant (*p* values < 0.05) for %R for all taxa and for nY for three of five taxa (Table 1, Figs 2–4).

The largest differences among treatments were observed in DNA extracts from *A. virens*. Tissues of *A. virens* stored in EDTA pH 10 yielded an average %R value of 61.3%, a value 8–11 times greater than those observed for tissues stored in EDTA pH 8 and 9 or 95% EtOH, but not significantly different than that observed from fresh tissue (Fig 2A). Similarly, tissues stored in EDTA pH 10 yielded a nY value of 0.07 μg DNA/mg tissue, 5–99 times greater than those of tissues stored in EDTA pH 8 and 9 or 95% EtOH, but not significantly different than that observed from fresh tissue (Fig 3A). Gel electrophoresis illustrates the qualitative

**Table 1. Summary of statistical model results.**

| | Taxa | Statistical test | Test statistic | df | p value |
|---|---|---|---|---|---|
| **%R** | *Alitta virens* | Friedman $\chi^2$ | $\chi^2 = 31.52$ | 4 | **<0.001** |
| | *Faxonius virilis* | Repeated Measures ANOVA | $F = 15.68$ | 4 | **<0.001** |
| | *Homarus americanus* | Friedman $\chi^2$ | $\chi^2 = 14.8$ | 4 | **0.005** |
| | *Scomberomorus maculatus* | Friedman $\chi^2$ | $\chi^2 = 27.36$ | 4 | **<0.001** |
| | *Mercenaria mercenaria* | Friedman $\chi^2$ | $\chi^2 = 22.16$ | 4 | **<0.001** |
| **nY** | *Alitta virens* | Friedman $\chi^2$ | $\chi^2 = 34.56$ | 4 | **<0.001** |
| | *Faxonius virilis* | Friedman $\chi^2$ | $\chi^2 = 15.7$ | 4 | **0.003** |
| | *Homarus americanus* | Friedman $\chi^2$ | $\chi^2 = 1.84$ | 4 | 0.765 |
| | *Scomberomorus maculatus* | Friedman $\chi^2$ | $\chi^2 = 18.96$ | 4 | **<0.001** |
| | *Mercenaria mercenaria* | Friedman $\chi^2$ | $\chi^2 = 9.36$ | 4 | 0.053 |

Percent high molecular weight DNA recovered (%R) and normalized high molecular weight DNA yield (nY) after 12 months of tissue storage. Significant $p$ values ($p < 0.05$) indicated with bold text; df, degrees of freedom.

differences among treatments, with distinct bands of HMW DNA evident in extracts from tissues preserved in EDTA pH 10 and in fresh tissue, as opposed to smears of low molecular weight DNA evident in other treatments (Fig 4A).

Extracts of tissues of *F. virilis* stored in EDTA pH 9 and 10 yielded average %R values of at least 59.6% (Fig 2B). These values were significantly greater than those observed for extracts stored in EDTA pH 8 and 95% EtOH and are as great or greater than for extracts from fresh tissue. Values of nY for all treatments were significantly greater than fresh tissue with the exception of EDTA pH 10, which was not significantly different from either fresh tissue or

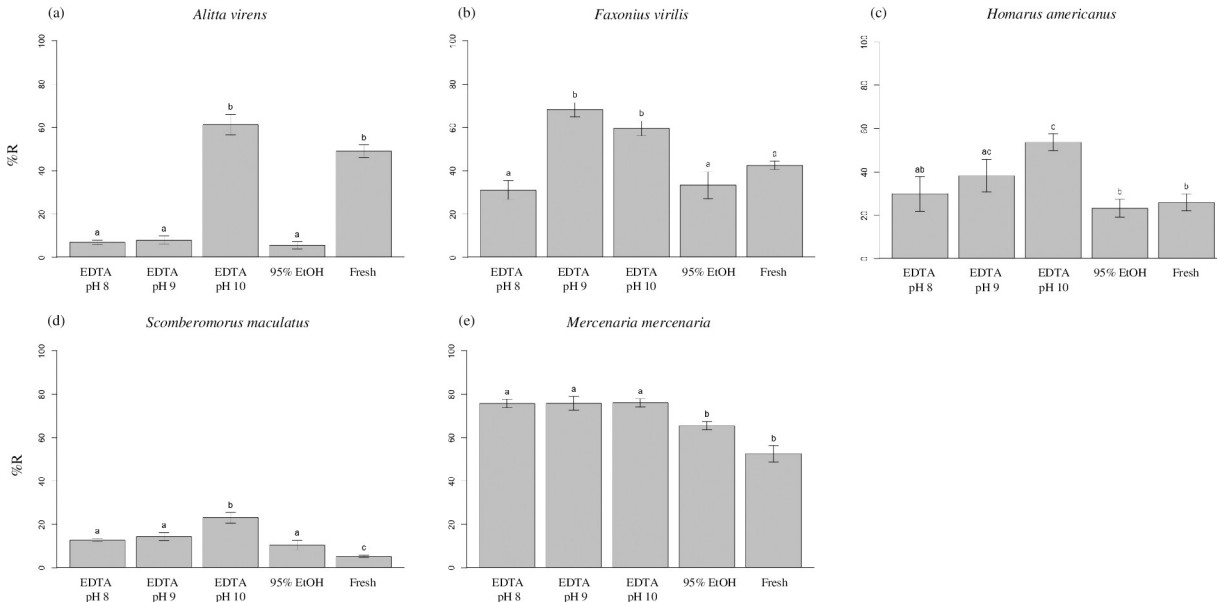

**Fig 2. Percent high molecular weight DNA recovered.** Average percent high molecular weight DNA recovered (%R) was determined for tissues of (a) *Alitta virens*, (b) *Faxonius virilis*, (c) *Homarus americanus*, (d) *Scomberomorus maculatus*, and (e) *Mercenaria mercenaria* that were either extracted immediately from fresh tissue or stored for 12 months at room temperature in 0.25 M EDTA adjusted to pH 8, 9, or 10, or 95% ethanol. Error bars represent standard error. Within this figure, treatments bearing different lower-case letters are significantly different at the Bonferroni-adjusted $p < 0.005$; matching lower-case letters indicate statistically indistinguishable treatments. 95% EtOH, 95% ethanol; Fresh, untreated tissue extracted immediately after dissection.

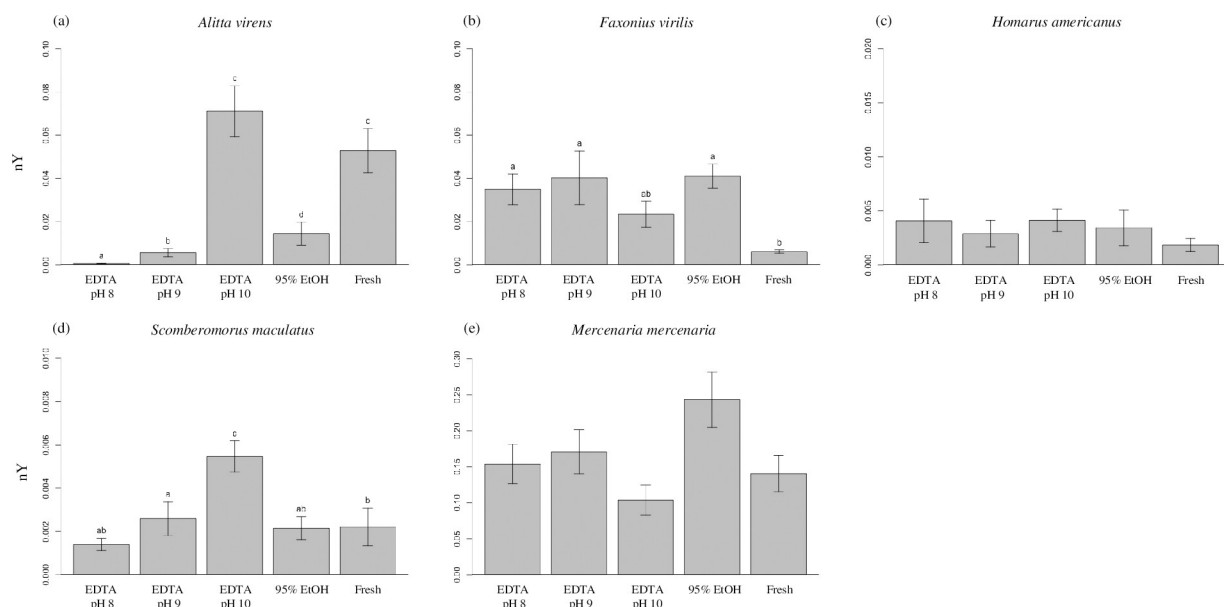

**Fig 3. Normalized high molecular weight yield.** Average normalized high molecular weight DNA yield (nY; μg DNA/mg tissue) was determined for tissues of (a) *Alitta virens*, (b) *Faxonius virilis*, (c) *Homarus americanus*, (d) *Scomberomorus maculatus*, and (e) *Mercenaria mercenaria* that were extracted either immediately from fresh tissue or stored for 12 months at room temperature in 0.25 M EDTA adjusted to pH 8, 9, or 10, or 95% ethanol. Error bars represent standard error. Within this figure, treatments bearing different lower-case letters are significantly different at the Bonferroni-adjusted $p < 0.005$; matching lower-case letters indicate statistically indistinguishable treatments; an absence of letters indicates no significant difference among all treatments in a given model. Note that y-axis scales differ among taxa. 95% EtOH, 95% ethanol; Fresh, untreated tissue extracted immediately after dissection.

from other preservative treatments (Fig 3B). Gel electrophoresis presents qualitative evidence for substantial DNA degradation in extracts of tissues stored in EDTA pH 8 and 95% EtOH (Fig 4B). Note that for two specimens, insufficient tissue was recovered for DNA extraction (each marked with an X in Fig 4B) and were excluded from further quantitative analyses.

Tissues of *H. americanus* stored in EDTA pH 10 yielded an average %R value of 53.7%, the highest value for this taxon (Fig 2C). This value is two-fold greater than that observed for 95% EtOH and is significantly greater than that for fresh tissue and all other treatments except EDTA pH 9. No significant differences in nY were observed among fresh tissue and all preservative treatments (Fig 3C). Gel electrophoresis reveals substantial degradation of DNA in extracts of all tissues stored in 95% EtOH and in some tissues stored in EDTA pH 8 and 9, while little evidence of DNA degradation is observed in extracts of tissues stored in EDTA pH 10 (Fig 4C). Although there was no significant difference in average %R between EDTA pH 9 and 10, the latter produced more consistent results among specimens.

Average values of %R for extracts of tissues of *S. maculatus* were lower than those observed for all other taxa, with no values exceeding 23.1% (Fig 2D). Note that specimens of *S. maculatus* were obtained postmortem and were maintained for an unknown period of time on ice before sampling. Nonetheless, both average %R and nY for extracts from tissues stored in EDTA pH 10 were significantly higher than for all other treatments and for fresh tissue (Fig 3D). Interestingly, average %R values for all treatments were significantly greater than for fresh tissue, while average nY values for EDTA pH 8 and 95% EtOH were not significantly different from fresh tissue. Gel electrophoresis results show substantial evidence of DNA degradation in fresh tissue and in all treatments except EDTA pH 9 and 10 (Fig 4D).

For *M. mercenaria*, the observed average %R values for extracts of fresh tissue and all preservative treatments were greater than 52.6% (Fig 2E). Extracts of tissues stored in EDTA pH

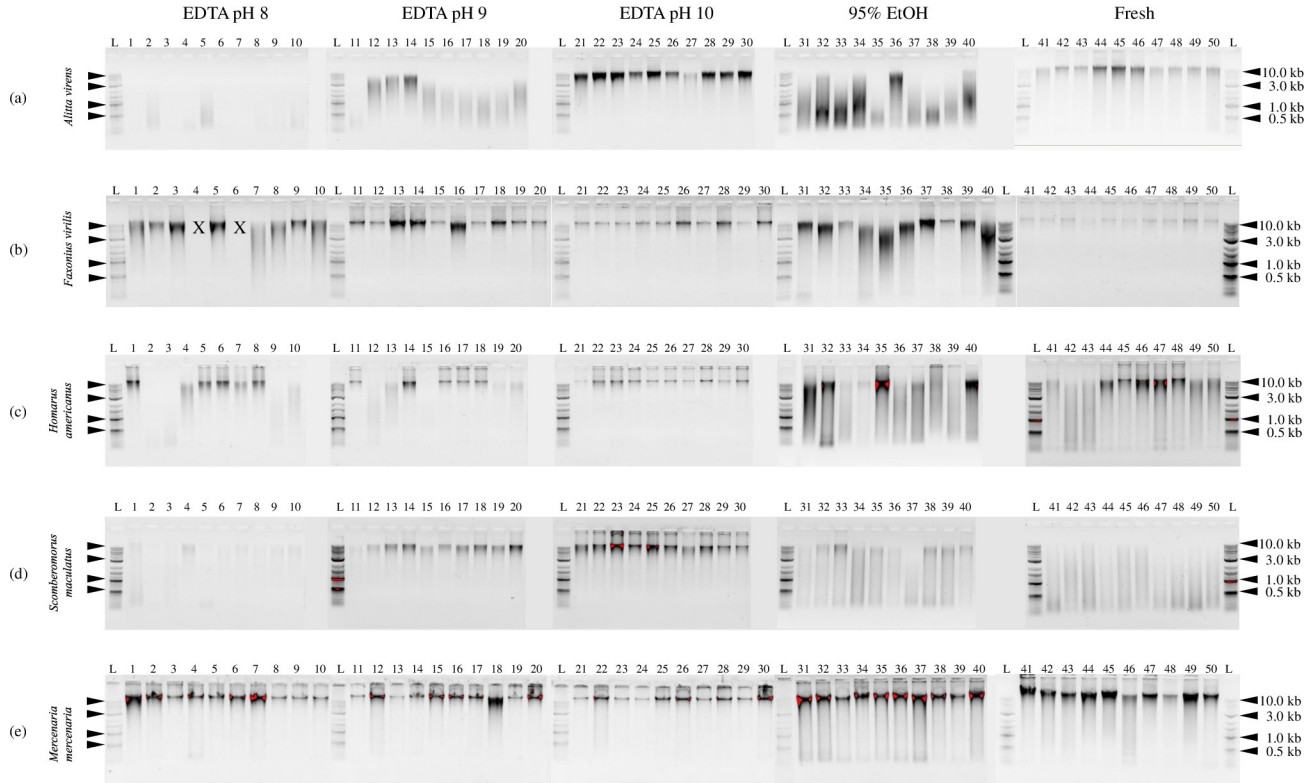

**Fig 4. Visualization of DNA extracts by gel electrophoresis.** DNA was extracted from (a) *Alitta virens*, (b) *Faxonius virilis*, (c) *Homarus americanus*, (d) *Scomberomorus maculatus*, and (e) *Mercenaria mercenaria* tissues after storage at room temperature for 12 months in 0.25 M EDTA pH 8 (lanes 1–10), 0.25 M EDTA pH 9 (lanes 11–20), 0.25 M EDTA pH 10 (lanes 21–30), and 95% ethanol (lanes 31–40). Fresh tissue extracts are shown in lanes 41–50. Lanes marked with an L contain 0.33 μL or 0.66 μL Quick Load Purple 1 kb Plus DNA Ladder (100 μg/mL; New England Biolabs; Ipswich, MA). Lanes marked with an X indicate samples that were not recovered. Specimens are presented in the same order across all treatments.

8, 9, and 10 yielded average %R values significantly greater than those for 95% EtOH and fresh tissue. No significant treatment effects were observed for average nY (Fig 3E). Results of gel electrophoresis also reveal qualitative evidence for greater DNA degradation in fresh tissue and tissues stored in 95% EtOH than for other treatments (Fig 4E).

The *COI* gene was amplified from DNA extracts of all fresh tissue samples and all tissue samples stored for 12 months in each preservative treatment. PCR products of the appropriate size were obtained from all extracts except one sample of *A. virens* stored in EDTA pH 8 and five samples stored in 95% EtOH: one sample each from *F. virilis* and *M. mercenaria* and three samples from *S. maculatus* (S1 Table). For all taxa, appropriately sized PCR amplicons were obtained from all fresh tissue samples and all tissues stored in EDTA pH 9 and 10.

For three specimens of each taxon, DNA sequencing was performed on PCR amplicons from fresh tissue and tissues stored for 12 months in EDTA pH 10 and 95% EtOH. All amplicons yielded bidirectional sequences of at least 500 bp with >97.5% of all base positions giving quality scores ≥Q20, with the exception of one sample of *S. maculatus* preserved in 95% EtOH that did not amplify. All sequences yielded species-level matches (>98% identity) to reference sequences in the BOLD database [28] for their respective species. In all cases, sequences obtained from fresh tissue were identical to those obtained from stored tissues, regardless of the storage treatment. Sequence identities to best matches in BOLD are reported in S2 Table.

## Discussion

Previous investigations have proposed that EDTA helps prevent degradation of HMW DNA in tissues by chelating divalent cations required as co-factors for most DNA-degrading enzymes [5, 6, 21]. The ability of EDTA to chelate divalent cations is known to increase exponentially with increasing pH over the range of pH 8 to pH 10 [23]. Therefore, increasing the pH of storage solutions containing EDTA may be expected to improve HMW DNA preservation. In this investigation, we demonstrate that in tissues of four of the five taxa examined, the percent recovery of HMW DNA (%R) improved with increasing pH of EDTA solutions. Specifically, after 12 months of room temperature storage, we observed significantly greater %R at pH 10 compared to pH 8 for these four taxa. Additionally, we observed significantly greater %R at pH 10 compared to pH 9 for two taxa and pH 9 compared to pH 8 for one taxon. Notably, in no case across all five taxa did %R decrease significantly with increasing pH. In two of five taxa, we also observed a significant improvement in normalized HMW DNA yield (nY) for tissues stored in EDTA pH 10 compared to pH 8 and 9. These results are consistent with the hypothesis that EDTA preserves HMW DNA by chelating divalent cations.

Interestingly, the effect of increased pH on preservation of HMW DNA in EDTA solutions was not consistent among all taxa examined. For example, in *M. mercenaria*, no difference in HMW DNA preservation was observed between EDTA solutions of increasing pH. Instead, high %R values (%R > 75%) were observed in all EDTA solutions regardless of pH. In contrast, for *A. virens*, only EDTA pH 10 was effective at preserving HMW DNA (%R > 61%). Low %R was observed for EDTA pH 8 and 9 (%R < 8%). These results can be explained if nucleases present in *M. mercenaria* are inactivated by EDTA for all tested pH values, whereas those present in *A. virens* can effectively compete with EDTA for binding divalent cations at lower pH values or are capable of functioning at lower concentrations of divalent cations.

Our results also suggest that EDTA may be an effective alternative to ethanol in many cases. Although ethanol-preserved tissues are widely used as a source of DNA for research, the effectiveness of ethanol may vary substantially among different taxa and tissues [33]. Indeed, many studies show low yields of HMW DNA from tissues preserved in ethanol during both short- and long-term storage [5, 7, 12, 14, 15, 34]. In this investigation, all EDTA solutions performed as well or better than 95% EtOH in preserving HMW DNA with respect to %R. Particularly, for EDTA pH 10, %R values were always significantly greater than those observed for 95% EtOH.

Surprisingly, %R values observed for tissues of all taxa preserved in EDTA pH 10 were not only always greater than those observed for EtOH-preserved tissues but were also as great or greater than those observed when DNA was extracted from fresh tissue. Additionally, for four taxa, tissues stored in EDTA pH 9 and for two taxa, tissues stored in EDTA pH 8, also yielded greater %R values than fresh tissue. This result may be explained if DNA degradation occurs during the lysis step of DNA extraction from fresh tissue but is inhibited in tissues that have been preserved with EDTA. Indeed, Lahiri and Schnabel [35] showed that the addition of EDTA to human blood reduced DNA degradation during the lysis step of DNA extraction. Our results suggest that even when fresh tissue is available for immediate DNA extraction, it may be preferable to first preserve tissues in EDTA pH 10.

Although %R is a good indicator of DNA preservation, many investigators have used PCR amplification success to evaluate the utility of DNA for downstream applications [5, 7, 9–11, 15, 36]. In our investigation, preservation of tissues in EDTA for pH values greater than 8 had no adverse effect on the amplification of the *COI* gene. PCR amplifications were successful from DNA extracted from all fresh tissue (50 of 50) and all tissues stored in EDTA pH 9 (50 of 50) and 10 (50 of 50). PCR amplification failed for DNA extracts from one tissue stored in

EDTA pH 8 (47 of 48) and 5 tissues stored in 95% EtOH (45 of 50). DNA sequencing was performed on PCR amplicons from three replicates of fresh tissue and tissues stored in EDTA pH 10 and 95% EtOH. In each case, *COI* sequences obtained from preserved tissues matched to those obtained from their fresh counterparts with 100% pairwise identity.

In conclusion, our results show that the performance of EDTA as a preservative of HMW DNA in many tissues improves with increasing pH. This is consistent with its increasing activity as a chelator of divalent cations as pH increases. Furthermore, we see no adverse effects on DNA quality or downstream applications when the pH of EDTA solutions is increased. Therefore, investigators may consider increasing the pH of EDTA-based preservative solutions to values greater than 8, especially for tissues or taxa where degradation of HMW DNA is known to be a problem.

## Supporting information

**S1 Table. Values for yield (μg), total normalized DNA yield (μg DNA/mg tissue), normalized high molecular weight DNA yield (nY; μg DNA/mg tissue), percent high molecular weight DNA recovered (%R), and *COI* PCR amplification success for each sample analyzed in this study.** Values are presented for tissues of *Alitta virens*, *Faxonius virilis*, *Homarus americanus*, *Scomberomorus maculatus*, and *Mercenaria mercenaria* that were either extracted immediately from fresh tissue or stored for 12 months in 0.25 M EDTA adjusted to pH 8, 9, or 10, or 95% ethanol. N/a indicates samples for which data were not collected; 95% EtOH, 95% ethanol; Fresh, untreated tissue extracted immediately after dissection; OGL, Ocean Genome Legacy Center specimen collection.
(PDF)

**S2 Table. DNA barcoding.** The barcode region of the mitochondrial cytochrome oxidase I (*COI*) gene was sequenced from three specimens of each taxon used in this study. The values listed are percent identities to the best match found in the Barcode of Life Data System (BOLD) database. Specimen IDs for both the Ocean Genome Legacy Center online catalog and best matches found in BOLD are presented. N/a indicates samples for which sequencing was not successful; OGL, Ocean Genome Legacy Center specimen collection.
(PDF)

**S1 Raw images. Raw agarose gel electrophoresis images for qualitative visualization of DNA fragment size distribution after 12 months.** DNA was extracted from (a) *Alitta virens*, (b) *Faxonius virilis*, (c) *Homarus americanus*, (d) *Scomberomorus maculatus*, and (e) *Mercenaria mercenaria* tissues after storage at room temperature for 12 months in 0.25 M EDTA pH 8 (lanes 1–10), 0.25 M EDTA pH 9 (lanes 11–20), 0.25 M EDTA pH 10 (lanes 21–30), and 95% ethanol (lanes 31–40). Fresh tissue extracts are shown in lanes 41–50. Lanes marked with an L contain 0.33 μL or 0.66 μL Quick Load Purple 1 kb Plus DNA Ladder (100 μg/mL; New England Biolabs; Ipswich, MA). Lanes marked with a red X indicate samples that were not recovered. Lanes marked with a black X indicate data that is not included in this investigation. Specimens are presented in the same order across all treatments.
(PDF)

## Acknowledgments

We would like to thank Bob and Eileen Matz, Don and Linda Comb, and many others who have supported the Ocean Genome Legacy Center. We also thank Emily Condon, the Northeastern University co-op program, the Marine Science Center community, and previous co-op students who have contributed to this research.

## Author Contributions

**Conceptualization:** Mia L. DeSanctis, Rosalia Falco, Hannah J. Appiah-Madson, Daniel L. Distel.

**Data curation:** Rosalia Falco, Hannah J. Appiah-Madson.

**Formal analysis:** Ella Messner, Rosalia Falco, Hannah J. Appiah-Madson.

**Funding acquisition:** Daniel L. Distel.

**Investigation:** Mia L. DeSanctis, Elizabeth A. Soranno, Ella Messner, Ziyu Wang, Rosalia Falco.

**Methodology:** Mia L. DeSanctis, Elizabeth A. Soranno, Elena M. Turner, Rosalia Falco, Hannah J. Appiah-Madson, Daniel L. Distel.

**Project administration:** Rosalia Falco, Hannah J. Appiah-Madson, Daniel L. Distel.

**Supervision:** Rosalia Falco, Hannah J. Appiah-Madson, Daniel L. Distel.

**Writing – original draft:** Mia L. DeSanctis, Ella Messner, Rosalia Falco, Hannah J. Appiah-Madson, Daniel L. Distel.

**Writing – review & editing:** Mia L. DeSanctis, Elizabeth A. Soranno, Ella Messner, Rosalia Falco, Hannah J. Appiah-Madson, Daniel L. Distel.

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
