## [Decision Letter · Decision Letter 0]

20 Dec 2022

PONE-D-22-14619Greater than pH 8: the pH dependence of EDTA as a preservative of high molecular weight DNA in biological samplePLOS ONE

Dear Dr. Distel,

Thank you for submitting your manuscript to PLOS ONE. After careful consideration, we feel that it has merit and it is very close to fully meeting PLOS ONE’s publication criteria as it currently stands. Therefore, we invite you to submit a revised version of the manuscript with minor revisions that address the questions raised during the review process, pertaining to the scope and justification of the work described. 

We look forward to receiving your revised manuscript.

Kind regards,

Marco Bonizzoni, Ph.D.

Academic Editor

PLOS ONE

Journal Requirements:

In your cover letter, please note whether your blot/gel image data are in Supporting Information or posted at a public data repository, provide the repository URL if relevant, and provide specific details as to which raw blot/gel images, if any, are not available. Email us at plosone@plos.org if you have any questions

Reviewers' comments:

Reviewer's Responses to Questions

**Comments to the Author**

1. Is the manuscript technically sound, and do the data support the conclusions?

Reviewer #1: Yes

Reviewer #2: Yes

2. Has the statistical analysis been performed appropriately and rigorously? 

Reviewer #1: Yes

Reviewer #2: Yes

3. Have the authors made all data underlying the findings in their manuscript fully available?

Reviewer #1: Yes

Reviewer #2: Yes

4. Is the manuscript presented in an intelligible fashion and written in standard English?

Reviewer #1: Yes

Reviewer #2: Yes

5. Review Comments to the Author

Reviewer #1: The manuscript is a great follow-up to a previous paper on DNA preservation using EDTA by the same laboratory (Sharpe et al., 2020, DESS deconstructed: Is EDTA solely responsible for protection of high molecular weight DNA in this common tissue preservative? PLOS ONE, https://doi.org/10.1371/journal.pone.0237356). In that study, the authors demonstrated that EDTA was the primary protector of high-molecular-weight DNA in the common preservative DESS. In this study they evaluate the effectiveness of EDTA as a DNA preservative at various pH’s. The results here are almost as striking and transformative as the previous study, notably so for difficult-to-extract taxa such as Alitta sp. Figure 4 is especially effective in demonstrating the utility of EDTA as a tissue preservative; the results shown in 4a are a bit astonishing.

This is an incredibly well-written and edited manuscript; I really have very few suggestions to make the body cleaner and more concise. The rare comments/questions are below.

Line 126, 127: Was the fresh tissue from the same individuals as the preserved tissue, which was extracted when the preserved tissues were put into storage? Alternatively, was the fresh tissue from new individuals, extracted with the preserved tissues after 12 months of storage? If the former, please briefly describe how the extracted DNA from fresh tissue was stored for those 12 months.

Line 163, 164: Why are the spaces in the sequences for primers LCO1490_t1 and HCO2198_t2 placed where they are? I initially thought it was marking where the primer ended and the M13 tail started, but it does not. I’m not sure if a space is needed in the sequence, but if so, I would think the best place would be between the amplicon primer and the M13 tail.

Line 211: You mention taking Absorbance ratios, and include the results in S1 Table, but never mention them at any other time. Were there any significant differences among ratios? Did you run any statistical tests? I find that Absorbance ratios are not very useful and are highly inconsistent in their ability to determine DNA quality, but if they are included, I would possibly mention their results somewhere. Evaluating them with S1 Table is not especially useful.

Table 1: I would mention that the p-values are Bonferroni-corrected in the table description.

Reviewer #2: dear authors,

This is an important study for DNA isolation works but please clearly explain the aims of study in introduction section. What do you want to find from these results? whether these finding can be industrialized or would be just experimentally?

Best wishes

6. PLOS authors have the option to publish the peer review history of their article (what does this mean?). If published, this will include your full peer review and any attached files.

Reviewer #1: No

Reviewer #2: **Yes: **Dr. Sakineh Abbasi

---

## [Author Response · Author response to Decision Letter 0]

6 Jan 2023

Response to Academic Editors and Reviewers

PONE-D-22-14619

Greater than pH 8: the pH dependence of EDTA as a preservative of high molecular weight DNA in biological sample

Response to Academic Editor

Journal Requirements:

Response: We have renamed the Supporting Information files to match PLOS ONE’s requirements.

Response: We have updated the Funding Information and the Financial Disclosure sections to ensure that they agree. We have also included an additional funder, the Francis Goelet Chartable Lead Trust, which was accidentally omitted from the original submission.

In your cover letter, please note whether your blot/gel image data are in Supporting Information or posted at a public data repository, provide the repository URL if relevant, and provide specific details as to which raw blot/gel images, if any, are not available. Email us at plosone@plos.orgif you have any questions

Response: We have included the uncropped gel images in the Supporting Information and have labeled them to indicate the regions of the gels that have been cropped to create Fig 4. All the data used in this investigation has been submitted in the Supporting Information section of our manuscript and so have not been submitted to a public data repository.

Response: The complete minimal data set required for this investigation is reported in the Supporting Information section of the manuscript. This includes S1 Table entitled Values for yield (μg), total normalized DNA yield (μg DNA/mg tissue), normalized high molecular weight DNA yield (nY; μg DNA/mg tissue), percent high molecular weight DNA recovered (%R), and COI PCR amplification success for each sample analyzed in this study, S2 Table entitled DNA barcoding, and S1 Fig entitled Raw agarose gel electrophoresis images for qualitative visualization of DNA fragment size distribution after 12 months.

Response: We have done so.

Response to Reviewer Comments

Reviewer #1: The manuscript is a great follow-up to a previous paper on DNA preservation using EDTA by the same laboratory (Sharpe et al., 2020, DESS deconstructed: Is EDTA solely responsible for protection of high molecular weight DNA in this common tissue preservative? PLOS ONE, https://doi.org/10.1371/journal.pone.0237356). In that study, the authors demonstrated that EDTA was the primary protector of high-molecular-weight DNA in the common preservative DESS. In this study they evaluate the effectiveness of EDTA as a DNA preservative at various pH’s. The results here are almost as striking and transformative as the previous study, notably so for difficult-to-extract taxa such as Alitta sp. Figure 4 is especially effective in demonstrating the utility of EDTA as a tissue preservative; the results shown in 4a are a bit astonishing.

This is an incredibly well-written and edited manuscript; I really have very few suggestions to make the body cleaner and more concise. The rare comments/questions are below.

Line 126, 127: Was the fresh tissue from the same individuals as the preserved tissue, which was extracted when the preserved tissues were put into storage? Alternatively, was the fresh tissue from new individuals, extracted with the preserved tissues after 12 months of storage? If the former, please briefly describe how the extracted DNA from fresh tissue was stored for those 12 months.

Response: We have modified the sentences to read as follows: DNA was immediately extracted from the fifth tissue sample of the same individuals without preservation, hereafter referred to as fresh tissue. The DNA from these fresh tissue samples was visualized immediately after extraction by gel electrophoresis and then stored at -80°C for 12 months, at which time DNA from all treatments, including fresh tissue, were analyzed concomitantly (lines 138-143).

Line 163, 164: Why are the spaces in the sequences for primers LCO1490_t1 and HCO2198_t2 placed where they are? I initially thought it was marking where the primer ended and the M13 tail started, but it does not. I’m not sure if a space is needed in the sequence, but if so, I would think the best place would be between the amplicon primer and the M13 tail.

Response: The extraneous spaces have been removed (lines 183 and 184).

Line 211: You mention taking Absorbance ratios, and include the results in S1 Table, but never mention them at any other time. Were there any significant differences among ratios? Did you run any statistical tests? I find that Absorbance ratios are not very useful and are highly inconsistent in their ability to determine DNA quality, but if they are included, I would possibly mention their results somewhere. Evaluating them with S1 Table is not especially useful.

Response: We agree with the reviewer’s assessment that the A260/A280 ratios reported in S1 Table are not very useful and that A260/A280 ratios are inconsistent in their ability to determine DNA quality. The A260/A280 ratio is an estimator of DNA purity, which was not examined in this investigation, and we used TapeStation data rather than A260 to estimate DNA concentration. For these reasons, we have removed A260/A280 ratios from the S1 Table and removed mention of them from the text, as suggested by the reviewer (removed from lines 178, 247, and 546).

Table 1: I would mention that the p-values are Bonferroni-corrected in the table description.

Response: The p values in Table 1 are not Bonferroni-corrected as they apply to single comparisons. However, we applied a Bonferroni correction to all post-hoc tests, e.g., in Figs 2 and 3. Here, the Bonferroni-corrected p value used for post-hoc tests is 0.005 rather than 0.05, reflecting the fact that ten pair-wise comparisons were made. We have corrected the legends of Figs 2 (line 263) and 3 (line 274) as follows: “Within this figure, treatments bearing different lower-case letters are significantly different at the Bonferroni-adjusted p < 0.005” to more accurately reflect the analysis performed. We have also clarified this point in the description of the statistical methods for parametric tests with the following text: “If the data were normal, a one-way repeated measures ANOVA and a Bonferroni-corrected Tukey post-hoc test was performed with the “nlme” package” (lines 237-239). 

Additionally, in reviewing the statistical analyses we noticed minor error. In one taxon, Faxonius virilis, for %R, we had not applied a Bonferroni correction. We have now done so. As a result, %R for this taxon is not different between EDTA pH 10 and fresh tissue. We have modified the text to reflect this (lines 303, 372-375, 401, and 403). These changes do not affect our conclusion that the performance of EDTA as a preservative of HMW DNA in many tissues improves with increasing pH. 

Reviewer #2: dear authors,

This is an important study for DNA isolation works but please clearly explain the aims of study in introduction section. What do you want to find from these results? whether these finding can be industrialized or would be just experimentally?

Response: We have added the following text to the introduction: “Our aim in this investigation is to determine whether the performance of EDTA as a preservative of DNA in tissue can be improved by increasing pH. If so, this simple modification could be of value for research and commercial applications” (lines 95-98).

---

## [Editor Report · Decision Letter 1]

10 Jan 2023

Greater than pH 8: the pH dependence of EDTA as a preservative of high molecular weight DNA in biological samples

PONE-D-22-14619R1

Dear Dr. Distel,

We’re pleased to inform you that your manuscript has been judged scientifically suitable for publication and will be formally accepted for publication once it meets all outstanding technical requirements.

Kind regards,

Marco Bonizzoni, Ph.D.

Academic Editor

PLOS ONE
---

## [Editor Report · Acceptance letter]

13 Jan 2023

PONE-D-22-14619R1 

Greater than pH 8: the pH dependence of EDTA as a preservative of high molecular weight DNA in biological samples 

Dear Dr. Distel:

I'm pleased to inform you that your manuscript has been deemed suitable for publication in PLOS ONE. Congratulations! Your manuscript is now with our production department. 

Kind regards, 

on behalf of

Dr. Marco Bonizzoni 

Academic Editor

PLOS ONE